# Hydration and Nephrolithiasis in Pediatric Populations: Specificities and Current Recommendations

**DOI:** 10.3390/nu15030728

**Published:** 2023-02-01

**Authors:** Maud Injeyan, Valeska Bidault, Justine Bacchetta, Aurélia Bertholet-Thomas

**Affiliations:** 1Reference Center of Renal Disease, Pediatric Nephrology, Rheumatology and Dermatology Unit, Filières ORKiD et ERKNet, Hôpital Femme Mère Enfant, 69500 Bron, France; 2Department of Pediatric Urologic and Visceral Surgery, Hôpital Femme Mère Enfant, 69500 Bron, France; 3Reference Center for Rare Diseases of Calcium and Phosphate Metabolism, Pediatric Nephrology, Rheumatology and Dermatology Unit, Filières OSCAR et BOND, Hôpital Femme Mère Enfant, 69500 Bron, France; 4INSERM Research Unit 1033, Pathophysiology of Bone Disease, Faculté de Médecine Lyon Est, Université de Lyon, Rue Guillaume Paradin, 69008 Lyon, France; 5Faculté de Médecine Lyon Est, Université Claude Bernard Lyon 1, 69001 Lyon, France

**Keywords:** nephrolithiasis, pediatrics, hydration

## Abstract

Renal lithiasis is less frequent in children than in adults; in pediatrics, lithiasis may be caused by genetic abnormalities, infections, and complex uropathies, but the association of urological and metabolic abnormalities is not uncommon. The aim of this study is to provide a synthesis of nephrolithiasis in children and to emphasize the role of hydration in its treatment. As an etiology is reported in 50% of cases, with a genetic origin in 10 to 20%, it is proposed to systematically perform a complete metabolic assessment after the first stone in a child. Recent data in the field reported increased incidence of pediatric urolithiasis notably for calcium oxalate stones. These changes in the epidemiology of stone components may be attributable to metabolic and environmental factors, where hydration seems to play a crucial role. In case of pediatric urolithiasis, whatever its cause, it is of utmost importance to increase water intake around 2 to 3 L/m^2^ per day on average. The objective is to obtain a urine density less than 1010 on a dipstick or below 300 mOsm/L, especially with the first morning urine. Some genetic diseases may even require a more active 24 h over-hydration, e.g., primary hyperoxaluria and cystinuria; in such cases naso-gastric tubes or G-tubes may be proposed. Tap water is adapted for children with urolithiasis, with limited ecological impact and low economical cost. For children with low calcium intake, the use of calcium-rich mineral waters may be discussed in some peculiar cases, even in case of urolithiasis. In contrast, sugar-sweetened beverages are not recommended. In conclusion, even if parents and patients sometimes have the feeling that physicians do not propose “fancy” therapeutic drugs, hydration and nutrition remain cornerstones of the management of pediatric urolithiasis.

## 1. Background

Even if renal lithiasis is less frequent in children than in adults [1,2], it represents 3% of the causes of kidney failure in adults. The other thing that differs from adults is the etiology. In pediatrics, genetic abnormalities can account for 10–20% of the causes, and complex uropathies and infections are also frequent causes. [3]. This is why it is essential to carry out a systematic assessment after the first stone in a child [4].

Even though urolithiasis is less frequent, it is a real concern for public health, because of the cost for the healthcare system on one hand, and because of the patients’ impaired quality of life on the other hand [5].

Diagnosis can be delayed especially in the youngest children with nonspecific symptoms [6]. Alpay et al. observed in a cohort of 162 children with urolithiasis that the most common presenting symptoms were flank pain or restlessness (25.3%) and hematuria (21.6%), followed by infections (16%), whereas 23.5% of the cases were detected incidentally during evaluation for other medical conditions [7].

Over the past two decades, an increase in the incidence of urolithiasis in pediatrics has been reported by different teams [8,9,10,11]. Robinson et al. have showed that the incidence of pediatric urology referrals for lithiasis was 1.77/100,000 person-years in north-west England, increasing annually by 13.6% between 2002 and 2015 [6].

The lithiasis process is secondary to urine oversaturation by crystals, depending on multifactorial parameters: insufficient water intake, over-excretion and/or excess of production of pro-lithogenic factors (such as calcium and oxalate), and lack of anti-lithogenic factors (such as citrate and magnesium). The composition of stones varies according to age and geographical area [3]. Lithiasis of infectious origin used to predominate in young boys, but recent data showed increased incidence of calcium oxalate stones in all age groups [6,8,9]. These changes in the epidemiology of stone components may be attributable to metabolic and environmental factors, hydration and nutrition playing a crucial role both to prevent and treat urolithiasis [10].

More generally in children, an association between better hydration and a more balanced diet has been shown, and chronic dehydration has been recently linked to obesity and malnutrition [12,13,14]. Thus, European and American guidelines have been published on the amount of water to be absorbed daily for children depending on age [15,16,17,18]. Unfortunately, a high proportion of children and teenagers are underhydrated, with a likely increased risk of kidney problems such as urinary tract infections, kidney stones, and acute kidney injury, in a global setting of other behaviors having negative impact on global health [19]. Indeed, poor nutrition, decreased physical activity, functional constipation, obesity, and other cardiometabolic risk factors are more and more seen even at the pediatric age [20].

The aim of this review is to discuss pediatric specificities of hydration in the context of urolithiasis.

## 2. Specificities of Urolithiasis in Pediatrics

### 2.1. Stone Compositions

Concerning stone composition, the main components of stones were calcium oxalate and calcium phosphate [1,4,8,9]. We recently reported an increase in the proportion of calcium oxalate stones and a decrease in the proportion of infectious stones in a cohort of 111 children with at least one stone analyzed by Fourier transformed infrared spectroscopy between 2013 and 2017 [3]. These changes in the epidemiology of stone components may be attributable to metabolic and environmental factors [3,8,21], keeping in mind that a significant part of pediatric urolithiasis is nevertheless due to genetic and metabolic causes. In contrast to adults in which urolithiasis is most often “idiopathic” or “diet-induced” [22], a cause is found in almost 50% of pediatric patients, metabolic disorders such as hypercalciuria, hypocitraturia, and hyperoxaluria being the more common [7].

The composition of stones depends on age: the proportion of calcium oxalate stones increase while carbapatite stones decrease with age. This can be partly explained by the evolution of etiologies by age group: indeed, metabolic and genetic disorders are more often found in infants before 1 year, infection in children between 1 and 5 years, and “idiopathic” causes after 6 years [3,6,7,21].

Data are more controversial for gender specificities [4], even though it is well accepted that gender distribution differs according to age. During the first decade, stone disease is more prevalent amongst boys, mainly due to a higher prevalence of urinary tract infections in boys during this life period. In contrast, during the second decade of life, girls are affected more often than boys [2]. Bergsland et al. reported that age and sex have a profound influence on urinary calcium and oxalate. Puberty is a time of rapid growth and hormonal changes, which could plausibly also affect stone pathogenesis [23].

### 2.2. From the Diagnostic Approach to Genetics

An etiology is reported in 30 to 80% of pediatric cases (average 50%) and is related to genetic abnormalities in 10–20% [3,7,24]. A complete metabolic assessment should be systematically performed after the first stone in a child, especially in the youngest ones [4]. Patients with stones present a two-fold increased risk of developing chronic kidney disease (CKD) in the long term compared to general populations [25]. It is crucial to provide an adequate diagnosis for these patients, since the prognosis of an orphan and severe inherited form of urolithiasis and nephrocalcinosis, namely primary hyperoxaluria type 1, has been recently dramatically modified by the onset of RNA-interfering therapies [26,27].

The first step is based on the search for factors pointing to a genetic origin: familial history of kidney stones and consanguinity, as well as active, bilateral, and early lithiasis. The second step is to identify favoring factors, notably urological abnormalities, chronic digestive diseases inducing malabsorption, drug intake, and/or history of infections. Table 1 summarizes the biological evaluation that should be performed when evaluating a child with a first episode of lithiasis [4,28]. In Table 2 there are pediatric references about plasmatic and urinary electrolytes [29]

When this complete metabolic evaluation is performed, we propose in Table 3 a classification that may help diagnosis and further genetic analysis and management of pediatric lithiasis.

### 2.3. Other Factors

Typically, in adults, hypercalciuria and hyperuricuria can be induced by a bad lifestyle. Indeed, high intake of sodium, animal proteins, calcium, and fructose increase calcium excretion and subsequently the incidence of calcium oxalate stones. Obesity is associated with risk factors contributing to the formation of lithiasis, such as lower urinary pH (due to insulin resistance) and increased excretion of calcium oxalate, uric acid, sodium, and phosphate [4]. The metabolic syndrome also leads to a defect of ammoniogenesis whilst an acidic urinary pH favors the precipitation of uric acid crystals [14]. While pediatricians observe an increased prevalence of overweight and obesity in children as well as metabolic syndrome in adolescents, there is also an increase of “nutritional” lithiasis at the pediatric age [3,22,30].

Complex uropathies malformations can favor lithiasis of infectious origin by stasis and hyperoxaluria (Ia) [31]. Infectious lithiasis such as struvite or carbapatite can be induced by an alkaline pHU. Indeed, struvite lithiasis (ammonium magnesium phosphate [AMP], IVc) is secondary to an infection with urea bacteria, in order of frequency: *Proteus mirabilis*, *Klebsiella pneumoniae*, *Staphylococcus aureus* or epidermidis, and *Pseudomonas* spp. A carbapatite lithiasis (IVa) with a carbonation rate higher than 15% also points to an infectious stone with urea bacteria; in such a case, an underlying metabolic factor is often associated.

Gut microbiota also seems to have an impact on the formation of nephrolithiasis. Gut microbiomes of children and adolescents with calcium oxalate kidney stone disease is less diverse [32]. In fact, the loss of bacteria producing butyrate and degrade oxalate is associated with perturbations of the microbiome and early-onset calcium oxalate kidney stone disease [33]. Denburg et al. have shown in a case-control study of 88 individuals aged 4–18 years a significantly less diverse gut microbiome in participants with lithiasis [32]. Many recent papers propose a model of lithogenesis prevention by using antibiotics, probiotics, and nutrition in children, but this remains to be further confirmed [34,35].

Last, secondary hyperoxaluria can be due to intestinal causes, malabsorption, low calcium intake, cystic fibrosis, shortened blood vessels, drugs, or toxins [36,37].

## 3. Overall Benefits of Hydration

### 3.1. Physiologically in Pediatrics

The water composition of the human body changes during the first years of life. In newborns, body water content at birth is approximately 75% of body mass. The relative water content of infants decreases rapidly throughout the first year of life to 60% and remains relatively stable throughout childhood until adolescence. In adults, water represents 50–60% of body mass [38]. Several studies have looked at tools for measuring water content in children. Bioelectrical impedance analysis and bioimpedance spectroscopy are commonly used for assessing body composition in children and adolescents. We can thus see an important difference of total body water between age, sex, and weight. On average, three-year-olds have total body water of 7 to 11 L depending on their weight and sex. At 10 years old, there is between 15 and 25 L depending on sex and age, and at 18 years old between 35 and 50 L for boys and between 25 and 38 L for girls [39,40,41]. A new method using isotope tracking (2 H) allows investigation of water turnover (water used by the body each day), confirming that age and body size were significantly associated with water turnover, as were physical activity, socioeconomic status, and environmental characteristics (altitude, air temperature) [42].

Water supports many functions essential in daily life, such as thermoregulation and waste elimination; it also serves as a carrier and a solvent for numerous metabolic reactions. In children, it is important to keep in mind physiological specificities of water balance, such as the progressive maturation of kidney function by around the age of two [43].

### 3.2. Drinking Habits

Children have a higher body water content relative to body mass and higher water requirements per unit of body weight as compared to adults [43,44]. Acquiring healthy drinking habits is important as early as possible, even in infancy. Parental influence on eating and drinking habits during childhood and school environment both play a crucial role in the acquisition of healthy consumption behaviors [44].

Results from nutritional surveys of Altaba et al., in 13 countries worldwide, showed a high proportion of children and adolescents at risk of inadequate water intake [45], leading to several kidney problems ranging from uncomplicated urinary tract infections to kidney stones, acute kidney injury, and chronic disorders with high mortality rates. Kavouras et al. showed that on average, 60 ± 24% (range 10–98%) of children from all over the world do not meet the daily water intake recommendations. These results support beliefs that children are not drinking enough water and that they are underhydrated [20].

A recent study found that promoting healthy hydration in elementary schools by increasing access to water through water fountains and water consumption classes was an effective strategy for reducing the risk of being overweight by 31% in the intervention group [12]. In this way, several studies are looking for biomarkers of inadequate hydration in children with the aim of large-scale prevention (urine color, urine osmolarity...) [19].

### 3.3. Overweight and Bad Eating Habits

There is a link between a better hydration and a more balanced diet in children [12], but other studies did not find any association between total water consumption and body mass index (BMI) in adults, children, and teenagers. The water content of food is a determinant of dietary energy density, which has been inversely associated with dietary quality. However, higher water intakes are suggestive of better diet quality [46].

However, downstream associations with low water intake, such as higher risk of hyperglycemia and CKD, are described in adults [47,48]. Low water intake in children is also associated with negative health behaviors (e.g., poor nutrition and decreased physical activity) and disease (e.g., postural tachycardia, urolithiasis, functional constipation, obesity, and other cardiometabolic risk factors) [20].

### 3.4. Hydration and Cognition

Dehydration has deleterious effects on cognitive performance, memory, and attention. Indeed, various studies show that dehydration of 1–2% of body mass reduces aerobic performance in prepubertal boys [49]. In addition, another study demonstrated that improving hydration levels through educational intervention resulted in a significant improvement in endurance performance in exercising children [50]. Last, a recent study showed that brain structure and function may be significantly affected in the dehydrated adolescent [51]. The benefits of improved hydration and acute water intake have been seen in attention, short term memory, delayed memory, visual search, and executive function, but not all studies have supported a relationship between hydration and cognition. The literature regarding how hydration affects cognition is still insufficient, and it would be beneficial to assess the effects of optimal and insufficient hydration on cognition and brain functions using a variety of laboratory and brain imaging techniques [52].

### 3.5. Water: Usual and Therapeutic Recommendations

Table 4 summarizes the recommended daily water intake depending on age from the different international guidelines, notably the Hydration for Heath initiative, National Health and Nutrition Examination Survey (NHANES), European Food Safety Authority (EFSA), and World Health Organization [15,16,17,18].

## 4. Place of Hydration in Nephrolithiasis Treatment

In the case of pediatric urolithiasis, whatever its cause, it is of utmost importance to increase water intake to around 2 to 3 L/m^2^ per day on average. The objective is to obtain a urine density less than 1010 on the dipstick or below 300 mOsm/L, especially on the first morning urine. Therefore, to be well distributed over 24 h, the patient should ingest a large quantity of water at bedtime and again during the night, taking advantage of the provoked nocturia. In infants and small children, a feeding or gastrostomy tube are sometimes required in the most severe forms of genetic lithiasis such as primary hyperoxaluria and cystinuria.

For several genetic abnormalities, it is necessary to increase the amount of water per day. In primary hyperoxaluria type 1 (PH1), it is classically recommended to increase fluid intake to over 3 L/m^2^ per day, distributed throughout 24 h, and to provide urine alkalinization and vitamin B6 to patients [53].

In cystinuria, the goal of treatment is to obtain solubilization of cystine excreted in urine with dietary measures (by controlling methionine and decreasing sodium intake to 1–1.5 mEq/kg/d) and to decrease cystine concentration by hyperdiuresis. It is recommended to maintain a high fluid intake to obtain cystine concentration under 250 mg/day in adults. In children, the target of urine specific gravity should be ≤1.005. Hyperhydration should be at least 2 L/1.73 m^2^, and the fluid intake should also ideally be distributed throughout the day and night. In order to increase cystine solubility, alkalinization using potassium citrate is a cornerstone of treatment for all patients [54,55].

Furthermore, if the patient had a surgery, it is even more important to respect supraphysiological hydration to avoid recurrences.

### 4.1. Focus on Medical Treatment and Genetic Diseases

In addition to symptomatic treatment, which includes hydration, some etiologies require specific and urgent management. Hence the importance of a full assessment at the time of the first episode in children.

Novel therapies based on RNA interference have recently emerged, and these therapies are a real game changer for managing patients with PH1, likely improving their quality of life and potentially preventing the need for transplantation [28,38,54].

For cystinuria, the basic treatment is to alkalinize the urine with citrate or potassium bicarbonate. When dietary measures and alkalinization seem to be well conducted, and if the lithiasis recurs, treatment with sulfhydryl derivatives can be proposed. These agents have various undesirable effects, and close systematic biological monitoring is necessary [55].

### 4.2. Which Water?

There is a very large amount of water available in Europe: tap water, prepackaged spring water, and prepackaged natural mineral water. Mineral waters are rich in calcium, rich in bicarbonate, and low in nitrates. Spring and tap waters are lower in sodium. In practice, we should harmonize our food intake with the types of beverages used, and low mineralized water is recommended for infants [56].

Tap water is adapted for children with urolithiasis, with limited ecological impact and low economical cost. For uric lithiasis, it is recommended to drink alkaline water or other bicarbonate rich water. Roux et al. showed that bicarbonate-enriched water had a positive action in bone metabolism [57].

Vitamin D and calcium deficiencies are common, and even more for children with overweight and obesity [58]. Many dietary components affect calcium absorption, but some mineral waters rich in calcium have absorption efficiency very close to dairy products [59]. In addition, for children, water with low mineral content is recommended (except in certain specific cases, especially low calcium intakes), but several studies are underway to assess impacts of ion content of drinking water and calcium oxalate crystal formation [60].

Children with nephrolithiasis should not be deprived of nutritional calcium, and calcium intake should be 80–100% of daily recommended intake [61]. Indeed, calcium deficiency may be deleterious for bone, and, in the past, association between hypercalciuria and low bone mineral density even at the pediatric age have been described [62]. Moreover, calcium deficiency is a risk factor of absorptive hyperoxaluria [63]. As such, we propose to follow the recent French guidelines on vitamin D and calcium intake in general pediatrics [59]. Of note, in such patients with nephrolithiasis, it may be wise to propose a daily vitamin D supplementation rather than intermittent supplementation [59,64]. The European Food Safety Authority (EFSA) proposed an overview of dietary reference values for calcium for children in Table 5 [61].

The consumption of sugar-sweetened beverages in children exceeds current recommendations. These drinks increase the risk for overweight/obesity and dental caries [65]. Moreover, several studies showed soda-like cola-type drinks with a high phosphate content are negatively associated with bone minerals and positively associated with bone fracture by a change of the calcium/phosphate ratio and the need to use bone tampons to counteract the acidic charge [66].

## 5. Conclusions

Nephrolithiasis is a common disease in children. Unlike adults, primary causes play an important role, and it is essential to look for genetic abnormalities, metabolic diseases, or malformations. Some genetic diseases require urgent treatment to improve the prognosis. However, we note an increase in the incidence of calcium oxalate stones and lithiasis of “idiopathic cause” in children over six years old, where hydration and nutrition play an essential role. Even if parents and patients sometimes have the feeling that physicians do not propose “fancy” therapeutic drugs, hydration and nutrition remain cornerstones of the management of pediatric urolithiasis. However, it is also important to keep in mind that water, which is closely related to nutrition, has an important role in the cognitive and motor development of the child and on overall health.

All in all, urolithiasis is a public health problem for which it is necessary to detect primary anomalies in children, but it is also necessary to prevent them through education about hydration and nutrition.

## Figures and Tables

**Table 1 nutrients-15-00728-t001:** Biological evaluation concerning every first episode of lithiasis in a childhood.

First line	**Blood Biology**	**Urinary Biology**	**Others**
	Ionogram: sodium, potassium, chloremia, creatinine, calcium, phosphate, bicarbonate, uric acid, magnesium, PTH, 25OHVitD	Density, osmolarity, calcium, phosphate, oxalate, cystine, citrate, magnesium, uric acid, creatinine, sodium, urea	Spectroscopy analysis: a carbonation rate (detected by infrared spectrometry) of less than 10% suggests lithiasis of metabolic origin (phosphate), whereas a carbonation rate of 15% or more points to infectious stones. Crystalluria (if available)
Additional explorations	When?	What?	Why?
	If hypercalciuria, or weddellite or brushite stones	Calcium load test	To show resorption or absorption hypercalciuria, or abnormalities of PTH regulation or Vitamin D metabolism
	If normal bicarbonate, hypocitraturia, normal or increased urinary pH, and carbapatite or weddellite stones	Acid load test	To demonstrate incomplete tubular acidosis

PTH: parathyroid hormone, 25OHVitD: 25 OH vitamin D.

**Table 2 nutrients-15-00728-t002:** Plasmatic and urinary electrolyte references in pediatrics.

	Age	Ratio Solute/Creatinine (95 ^e^ per) mmol/mmol mg/mg		Urinary 24 h (d)
Calcium	0–6 months 7–12 months 1–3 years 3–5 years 5–7 years >7 years	<2 <1.5 <1.5 <1.1 <0.8 <0.6	<0.8 <0.6 <0.53 <0.39 <0.28 <0.21	<0.1 mmol/kg/d (<4 mg/kg/d)
Oxalate	0–6 months 7–24 months 2–5 years 5–14 years >16 years	<0.36 <0.17 <0.10 <0.08 <0.04	<0.26 <0.14 <0.08 <0.06 <0.03	<0.5 mmol/1.73 m^2^/d (<45 mg/1.73 m^2^/d)
Citrate	0–5 years >5 years	>0.25 >0.15	>0.42 >0.25	M: > 1.9 mmol/1.73 m^2^/d (>365 mg/1.73 m^2^/d) F: > 1.6 mmol/1.73 m^2^/d (> 310 mg/1.73 m^2^/d)
Uric acid	<1 years 1–3 years 3–5 years 5–10 years >10 years	<1.5 <1.3 <1.0 <0.6 <0.4	<2.2 <1.9 <1.5 <0.9 <0.6	<70 µmol/kg/d (<1.3 mg/kg/d) <65 µmol/kg/d (<1.1 mg/kg/d) <65 µmol/kg/d (<1.1 mg/kg/d) <55 µmol/kg/d (<0.9 mg/kg/d) <55 µmol/kg/d (<0.9 mg/kg/d)
Magnesium	>2 years	>0.63	>0.13	>0.04 mmol/kg/d (>0.8 mg/kg/d)
Cystine	<10 years >10 years Adult	<12 <12 <12	<0.07	<55 µmol/1.73 m^2^/d (<13 mg/1.73 m^2^/d) <200 µmol/1.73 m^2^/d (<48 mg/1.73 m^2^/d) <250 µmol/1.73 m^2^/d (<60 mg/1.73 m^2^/d)
Creatinine	3–5 years 6–8 years 14–18 years			12–20 mg/d 15–25 mg/d M: 18–27 mg/d F: 17–24 mg/d
Phosphore		mmol/L		TmP/GFR (mmol/L) urinary
	1–3 years 3–5 years 5–7 years 7–9 years 9–11 years 11–13 years 13–16 years 16–19 years	1.38–2.19 1.38–2.19 1.33–1.92 1.33–1.92 1.33–1.92 1.33–1.92 F: 1.02–1.79 M: 1.14–1.99 0.95–1.62		1.53 (1.13–1.92) 1.47 (1.19–1.74) 1.42 (1.13–1.70) 1.40 (1.11–1.69) 1.41 (1.14–1.67) 1.41 (1.14–1.67) F: 1.24 (0.87–1.60) M: 1.34 (0.98–1.69) F: 1.12 (0.77–1.46) M: 1.16 (0.71–1.61)

**Table 3 nutrients-15-00728-t003:** Proposed classification of hereditary renal lithiasis in children.

	Biology	Etiology	Lithiasis	Genetic
Hypercalciuria	Normal Ca Normal PTH Normal Ca/creat (U) After calcium loading, adapted PTH braking and Delta Ca/creat (U) >0.05 mmol/mmol	Anomaly VitD metabolism	Weddellite (IIa)/carbapatite (IVa1)/Brushite (Ivd)	Inhibitory mutations of 24 hydroxylase (*CYP24A1* gene)
		Familial hyperparathyroidism		*MEN1*
				*HRPT2*
				*Ca Sr* genes
	Normal or high Ca High PTH High Ca/creat(U) After calcium loading, high PTH and delta Ca(U)/creat(U) < 0.05 mmol/L	Anomaly tubular reabsorption Ph		Gene *Npt2a, Npt2c*
				Gene *NHERF1*
Hyperoxaluria	Oxalate/creat U increased	Type 1	Whewellite (Ia/Ic)	*AGXT*
		Type 2		*GRHPR*
		Type 3		*HOGA1*
Tubular acidosis, Uric acid lithiasis	Acide pH U Urinary uric acide/creatinuria > 1.5 mmol/mmol (<2 years), >0.4mmol/mmol (>10 years)	Hyperuciemia	Type III	*HRPT* (Lesh Nyhan syndrom) X-linked recessive
		PRPPS		X-linked recessive
		APRT		Autosomal recessive
Cystinuria	Alkaline pH Cystinuria increased	Defects in the reabsorption of dibasic amino acids	Type V	*SLC3A1* (type A), *SLC7A9* (type B)

Ca: calcemia, PTH parathyroid hormone, creat: creatininemia, VitD: vitamin D.

**Table 4 nutrients-15-00728-t004:** Recommendation guidelines from IoM (Institute of Medicine) 2004, EFSA 0, (European Food Safety Authority), WHO (World Health Organization) 2005.

	IoM 2004	EFSA 2010	WHO 2003, 2005
1–2 years	1.3 L/d	1.1 to 1.2 L/d	1.0 L/d
2–3 years		1.3 L/d	
4–8 years	1.7 L/d	1.6 L/d	Female: 2.2 L/d Male: 2.9 L/d
9–13 years	Female: 2.1 L/d Male: 2.4 L/d	Female: 1.9 L/d Male: 2.1 L/d	
14–18 years	Female: 2.3 L/d Male: 3.3 L/d	Female: 2.0 L/d Male: 2.5 L/d	
>18 years	Female: 2.7 L/d Male: 3.7 L/d		

**Table 5 nutrients-15-00728-t005:** Overview of Dietary Reference Values for calcium for children according to EFSA journal.

	D-A-CH (2015)	NCM (2014)	IOM (2011)	WHO/FAO (2004)	Afssa (2001)	NL (2000)	SCF (19934)	DH (1991)
Age (months) PRI (mg/d)	4–12 330	6–11 540	6–12 260	7–12 400		6–11 450	6–11 400	0–12 525
Age (years) PRI (mg/d)	1–4 600	1–5 600	1–3 700	1–3 500	1–3 500	1–3 500	1–3 400	1–3 350
Age (years) PRI (mg/d)	4–7 750	6–9 700	4–8 1000	4–6 600	4–6 600	4–8 700	4–6 450	4–6 450
Age (years) PRI (mg/d)	7–10 900	10–17 900	9–18 1300	7–9 700	7–9 900	9–18 1200 (M) 1100 (F)	7–10 550	7–10 550
Age (years) PRI (mg/d)	10–13 1100			10–18 1300	10–19 1200		11–17 1000 (M) 800 (F)	11–18 1000 (M) 800 (F)
Age (years) PRI (mg/d)	13–19 1200							

PRI: population reference intake, M: males, F: females.

## Data Availability

All data supporting the findings of this article are included in the manuscript.

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
