# Peer review of "Hydration and Nephrolithiasis in Pediatric Populations: Specificities and Current Recommendations"

_nutrients, 2023, doi:10.3390/nu15030728_

Round 1

Reviewer 1 Report

General comment

The manuscript entitled “Hydration and nephrolithiasis in pediatric populations: specificities and current recommendations aims to summarize the specificities of hydration in the pediatric population, in relation to urolithiasis. Albeit the manuscript is interesting and summarizes an interesting topic, the manuscript requires several corrections before proceeding to the publication. In particular, in addition to the corrections related to typos and English grammar, the construction of the manuscript should be revised and some sections expanded. The suggested corrections are reported below in detail.

Major issues

ABSTRACT

The abstract, even if not structured should report the aim of the study, the main findings and the conclusions.

BACKGROUND

The introduction should be improved, reporting the epidemiological data of urolithiasis, its incidence in pediatric populations and the potential risk associated with this disease.

SPECIFICITIES OF UROLITHIASIS IN PEDIATRICS

60: According to the introduction, you could briefly report these data in the introduction and utilize this section to report the specificities of reported epidemiological data, as you wrote in the first part of this paragraph.

118: this section regarding lifestyle and other factors should be expanded as, apart from the genetic abnormalities, they represent the main reason for urolithiasis.

OVERALL BENEFITS OF HYDRATION

149-158: This is too generical and simplistic

TREATMENT OF LITHIASIS

I would suggest you to postpone the recommendation for genetic abnormalities at the end of the section and highlight the other subparagraphs

227: to this regard also see: https://doi.org/10.3390/cryst11121507

Minor issues

BACKGROUND

33-37: This is redundant with the abstract, please expand and change the sentences.

41: Delete etc, report mechanism or avoid generalizations.

SPECIFICITIES OF UROLITHIASIS IN PEDIATRICS

73: I would focus this section on stone compositions

129: Check typos

Author Response

The manuscript entitled “Hydration and nephrolithiasis in pediatric populations: specificities and current recommendations aims to summarize the specificities of hydration in the pediatric population, in relation to urolithiasis. Albeit the manuscript is interesting and summarizes an interesting topic, the manuscript requires several corrections before proceeding to the publication. In particular, in addition to the corrections related to typos and English grammar, the construction of the manuscript should be revised and some sections expanded. The suggested corrections are reported below in detail.

Thanks for having taking time to review our manuscript and to propose relevant comments to improve it.

  • ABSTRACT : The abstract, even if not structured should report the aim of the study, the main findings and the conclusions.

Thanks for this comment. In fact, we have structured our abstract and added a sentence explaining the aim of the review. You can see the new sentence line 18-19.

  • BACKGROUND :

Major issue : The introduction should be improved, reporting the epidemiological data of urolithiasis, its incidence in pediatric populations and the potential risk associated with this disease.

Thank you for this comment. To improve our background, we have reported the epidemiology part of the part 2.1 to the introduction (l 44-55).

Minor issue : 33-37: This is redundant with the abstract, please expand and change the sentences.  41: Delete etc, report mechanism or avoid generalizations.

Thank you for these comments, so we have changed the sentences of lines 39 to 43. “etc” was deleted on line 60.

  • SPECIFICITIES OF UROLITHIASIS IN PEDIATRICS

60: According to the introduction, you could briefly report these data in the introduction and utilize this section to report the specificities of reported epidemiological data, as you wrote in the first part of this paragraph.

73: I would focus this section on stone compositions    

Thank you for this comment, that what’s we did for the part 2.1, line 80. We also have changed the title to “stone compositions”.

118: this section regarding lifestyle and other factors should be expanded as, apart from the genetic abnormalities, they represent the main reason for urolithiasis.

Thank you for the comment, this review focused in the pediatric population seems to be one part of a review dedicated to lithiasis in adults and children. The adult topic will be developped in another part.

129: Check typos

Thank you for this comment, there were mistakes in english language and typos. Line 144

  • OVERALL BENEFITS OF HYDRATION : 149-158: This is too generical and simplistic

Thanks for this comment. We have added information on tools measuring water content, what is commonly done and what is new. Lines 168 -174

  • TREATMENT OF LITHIASIS : I would suggest you to postpone the recommendation for genetic abnormalities at the end of the section and highlight the other subparagraphs.

Thank you for this comment. We have structured this part, called now “place of hydration in nephrolithiasis treatment” with a first section of hydration whatever the cause, and a second part with hydration with genetic abnormalities. Lines 229 and 238. And we have separated the subsection of medical treatment from this section, so this is a new part called “focus in medical treatment and genetic disease, line 252

227: to this regard also see: https://doi.org/10.3390/cryst11121507

Thank you for this comment. We have added this reference in the section of “wich water”, this article is very important point. This data doesn’t seem to be according with pediatric but we mentioned it, line 275-277.

Reviewer 2 Report

Dear authors, thank you for your manuscript. It's a good review that can be useful for all the specialist and also for a newbie. Have you some references about the urinary electrolytes in the first year of life?   Can you add to your manuscript the normal range values (for age) of urinary electrolytes and others as for example Ca(U) /Creat (U)? Thanks

Author Response

Dear authors, thank you for your manuscript. It's a good review that can be useful for all the specialist and also for a newbie.

We would like to thank the reviewer for his overall positive appreciation of our manuscript.

Have you some references about the urinary electrolytes in the first year of life?   Can you add to your manuscript the normal range values (for age) of urinary electrolytes and others as for example Ca(U) /Creat (U)? Thanks

Thanks for this very relevant comment. In fact, we have added in table 1b, pediatrics references about plasmatic and urinary electrolytes. Line 119

Reviewer 3 Report

The paper looks fine. Please check lines 129- '130 and 139- 143 for minor English language corrections. Could you develop section 3.4? In what concerns section 4, I would rename it 'Treatment of lithiasis by hydration' and add some information about the medical treatment of urolithiasis in children. Any recommendations regarding the place of hydration in 'surgical' pediatric lithiasis?  

Author Response

The paper looks fine.

Thanks for having taking time to review our manuscript and to propose relevant comments to improve it.

Please check lines 129- '130  and 139- 143  for minor English language corrections.

Thank you for this comment, We have changed this, lines 144 and 154-155

 Could you develop section 3.4?

Thank you for this comment. In fact we have developed this part and this is much more interesting, lines 168-175.

In what concerns section 4, I would rename it 'Treatment of lithiasis by hydration' and add some information about the medical treatment of urolithiasis in children.

Thank you for this comment, we have added some information about the medical treatment and done a part called “focus in medical treatment and genetic diseases”. You can see lines 256 to 260.

Any recommendations regarding the place of hydration in 'surgical' pediatric lithiasis?

Thank you for this comment, we have added a sentence explaining the importance to respect hydration whether the child have had a surgery. Lines 250-251.

Round 2

Reviewer 1 Report

The authors improved the manuscript accordingly to previous suggestions.